# The Genomic Signatures of Linitis Plastica Signal the Entrance into a New Era: Novel Approaches for Diagnosis and Treatment

**DOI:** 10.3390/ijms241914680

**Published:** 2023-09-28

**Authors:** Grigorios Christodoulidis, Konstantinos Eleftherios Koumarelas, Marina Nektaria Kouliou, Maria Samara, Eleni Thodou, Dimitris Zacharoulis

**Affiliations:** 1Department of General Surgery, University Hospital of Larissa, University of Thessaly, Biopolis Campus, 41110 Larissa, Greece; kostaskoumarelas@gmail.com (K.E.K.); marinakouliou@gmail.com (M.N.K.); zachadim@yahoo.com (D.Z.); 2Department of Pathology, Faculty of Medicine, School of Health Sciences, University of Thessaly, Biopolis Campus, 41110 Larissa, Greece; msamar@med.uth.gr (M.S.); ethodou@uth.gr (E.T.)

**Keywords:** Linitis Plastica, Borrmann type IV, oncogenes in LP, biomarkers, targeting agents, surgery

## Abstract

Linitis Plastica (LP) is a rare and aggressive tumor with a distinctive development pattern, leading to the infiltration of the gastric wall, the thickening of the gastric folds and a “leather bottle appearance”. LP is an extremely heterogeneous tumor caused by mutations in oncogenic and tumor suppressive genes, as well as molecular pathways, along with mutations in stromal cells and proteins related to tight junctions. Elucidating the molecular background of tumorigenesis and clarifying the correlation between cancerous cells and stromal cells are crucial steps toward discovering novel diagnostic methods, biomarkers and therapeutic targets/agents. Surgery plays a pivotal role in LP management, serving both as a palliative and curative procedure. In this comprehensive review, we aim to present all recent data on the molecular background of LP and the novel approaches to its management.

## 1. Introduction

Gastric cancer (GC) ranks as the fifth most common malignancy worldwide, with approximately 1.1 million new cases annually. It stands out as having one of the highest mortality rates with the third most deaths [1,2,3,4]. Linitis Plastica (LP), a subtype of gastric cancer, constitutes 10% of all GC cases [2,5,6]. First observed in the 16th and 17th centuries, it was formally described in 1859 by William Brinton [7]. Macroscopically, LP is characterized by the thickening of the gastric folds due to accumulation of fibrous tissue, often referred to as the “Leather-bottle stomach” [3,5,7,8]. The term “Linitis” derives from its “linen-like” macroscopical image. LP originates from the fundic gland cells and infiltrates the entire stomach with a desmoplastic growth pattern, unlike the development of a solid tumor. Tumor cells are dispersed within the thickened fibrous tissue, extending into the submucosal and serous layer [5,7,9,10,11]. In 91.1% of cases, LP presents as a poorly differentiated adenocarcinoma, and in 77.7% of cases, it is often histologically linked with signet ring cells [8]. However, diagnosing LP is a complex and challenging endeavor, due to the infiltrative nature of the tumor, which often leads to false-negative biopsy results. Additionally, the absence of a standardized classification and definition protocol for LP further compounds the diagnostic difficulties [5,6,7]. While LP was initially classified by Borrmann, as early as 1926, as type IV, it has been labeled differently in subsequent classifications, such as diffuse gastric cancer in Lauren’s classification, as scirrhous cancer in the Japanese classification, and more recently, as poorly cohesive cancer, in the WHO classification; thus, there is still lack of standard criteria for the macroscopical and microscopical definite diagnosis of LP, which is indicative of its vast heterogeneity [7,11]. In the currently available literature, LP has also been classified as Borrmann type IV cancer, signet ring cell cancer, Lauren’s diffuse cancer and scirrhous cancer. However, these terms do not always refer to the same cancer type. LP predominantly affects women and younger individuals, distinguishing it from other types of GC [8,11]. Its aggressive behavior, stemming from its unique features, places the 5-year overall survival (OS) at 5%, while non-LP GC patients have a median 5-year OS of 36%, respectively [8,9,12,13]. Only a small percentage of LP patients are diagnosed at an early stage and are consequently candidates for R0 resection. However, even for this group of patients, the recurrence rate can soar as high as 60 to 70%, due to the elevated incidence of peritoneal carcinomatosis [11]. These challenges underscore the imperative the need for further research into novel approaches, as well as the molecular background of LP, to improve patient survival rates. In this systematic review, we aim to examine the currently available literature and present the latest data concerning the molecular background of LP, the role of epigenetics in oncogenesis, biomarkers for early LP detection, as well as the critical role of surgery in enhancing patient survival rates.

## 2. Methods

PubMed, Cochrane Library, Medline Scopus-clinical trial register, and Web of Science databases were initially searched by the authors to retrieve studies reporting data on Linitis Plastica from 2017 until present day. The following Medical Subject Heading [MeSH] terms alone or matched by the logical operators “OR” or “AND” were used: “Linitis Plastica”, “Scirrhous cancer”, “Gastric cancer”, “Lauren diffuse type”, “Borrmann type IV cancer”, “Oncogenes”, “Molecular background”, “Tumorigenesis”, “Targeting agents”, “Biomarkers”, “Novel therapies”, “Novel targets”, “Role of surgery”. Old, repetitive, and non-English studies were excluded. After an initial title screening, each relevant article was subsequently reviewed, and 54 representative scientific papers were finally selected.

## 3. Molecular Background (Oncogenes, Drivers, Pathways)

Scirrhous cancer, like all other cancers, follows a common developmental pattern characterized by driver mutations, loss of function in tumor suppressor genes and deviation in apoptosis. However, these mutations in scirrhous cancer result in a distinct invasive and metastatic pattern, significantly decreasing any chance of survival. Determining is the role of stromal mutations and the influence of stromal cells to the cancerous ones [2,7,10,14]. Cancerous cells in LP often develop in hypoxic conditions, activating HIF-1a (Hypoxia Induced Factor), which, in turn, induces the production of ANGPTL4 [15]. ANGPTL4 promotes tumor growth, up-regulates *c-Myc*, down-regulates *p27* and leads to a resistance in anoikis (cell death) by activating FAK, PI3K-Akt and ERK signaling pathways. The expression of all these genes is directly associated with peritoneal metastasis [15]. Liu et al., in their cohort study, observed that one of the most significant factors in the proliferation of cancerous cells, as well as in their communication with the stromal ones, is the Hippo pathway. The hippo pathway consists of tumor suppressor genes with two of these key genes (*FAT3* and *PTPN14*) being highly mutated in LP. This highlights the crucial role of the Hippo pathway in scirrhous cancer [3]. Three additional pathways have been identified to possess a significant role in tumorigenesis. The KEGG pathway is correlated with the extracellular matrix and contributes to pathological stromal stiffness. At the same time, the *OR51B5* gene’s up-regulation increases collagen biosynthesis [3,16]. The PI3KT-AKT pathway is up-regulated in LP, while the AMPK pathway is down-regulated, both contributing to cell growth due to their “oncogene-like functioning” [3]. Huang et al., in their cohort, report that the most frequently traced mutations among LP patients were *RELN, CDH1* and *ARID1A*, with occurrence rates of 76%, 65% and 40%, respectively [9]. On the other hand, Liu et al. demonstrate that the most common mutations affect the *TTN* gene, followed by *TP53, MUC16, CDH1, LRP1B, CYNE1* and *ARID1A*. These findings demonstrate the increased heterogeneity of LP, which complicates the development of specific therapeutic agents [3]. As mentioned above, another factor promoting tumorigenesis, particularly the invasive and metastatic pattern of LP, is the mutations of tight junction genes. Claudin 18.2 is among the most critical proteins involved in tight junctions and is linked to the epithelial–mesenchymal transition, the diffuse invasive pattern and metastasis. However, no significant difference was found in the OS among patients with positive *CLDN18* gene and those with the negative one [17]. The *CDH1* mutation is present in approximately 20% of LP patients, making it one of the most common mutations linked with diffuse gastric cancer (DGC). The methylation of the *CDH1* gene down-regulates the expression of E-cadherin [3]. E-cadherin and Desmoglein-2 mutations also play a crucial role in the invasive pattern of LP [10]. The down-regulation of E-cadherin expression is additionally derived from the low expression of miRNA-200c, which results in the over-expression of ZEB1 and, subsequently, a depletion of E-cadherin [2]. Scirrhous cancer exhibits a distinctive pattern of tumorigenesis, characterized by increased crosstalk among the cancerous and stromal cells. Stromal fibroblasts secrete a variety of molecules that interfere with tumor growth and metastasis. Kasai et al. suggested that the hepatocyte growth factor (HGF) and the MET receptor play major roles and are the main causes of peritoneal carcinomatosis and ascites, while Miki et al. established that FG7, secreted by stromal fibroblasts, promotes tumor progression and increases the invasion and migration of cancerous cells through the THBS-1 pathway in cancer cells with FGFR2 amplification [10,14,18]. FGFR2 amplification is present in approximately 5 to 10% of GC cases, with a higher percentage in scirrhous cancer. FGF9 exerts a paracrine action and activates MMP7, leading to extracellular matrix degradation, a diffuse invasive pattern and the promotion of apoptosis evasion [10]. Yashiro et al. observed that Asprorine (ASPN), a proteoglycan primarily traced in the fibroblasts, is directly correlated with LP, worsening survival and prognosis (*p* < 0.01) [19]. Last but not least, CD 63, an exosome associated with extracellular communication, possesses a significant role in tumor prognosis, due to its correlation with scirrhous cancer, patients aged over 65, a tumor depth of T3 or T4, lymphatic metastasis and a tumor size of over 5 cm [18]. Loss of function in the *STK11* or *LKB1* tumor suppressor genes, as well as *CD44*-*IGF1R* fusion, or the up-regulation of *NOS3* and *ARHGAP4* also leads to LP, increases nodal involvement and promotes epithelial mesenchymal transition through the MAPK pathway [16,20,21]. The up-regulation of *ARHGAP4* activates MMP2 and MMP9 metalloproteinases and decreases the expression of E-Cadherin and Claudin-1, thereby promoting the distinctive invasive pattern of LP [16]. We include a table outlining the main oncogenes in LP (Table 1).

## 4. Biomarkers

As mentioned previously, LP is characterized by a diffuse infiltration pattern that originates from the submucosal layer. Simultaneously, reactive fibrosis thickens the gastric folds. However, despite this being a distinct feature of the disease, up to 30% of patients do not exhibit the same endoscopic appearance. LP can sometimes present as a non-specific gastritis or even as a normal mucosa. The “flat type” of LP may also be mistaken for atrophic gastritis. Endoscopic ultrasound (EUS) with biopsy is considered the gold standard when diagnosing gastrointestinal malignancies. However, an increased percentage of false negative biopsies is observed, due to the mucosal layer, which usually remains intact; the reactive fibrosis; and scattered cancer cells originating from the diffuse pattern [6,7]. Thus, the diagnostic procedure is often challenging, with false-negative biopsies occurring in approximately 30 to 55% of patients. A recent meta-analysis of four retrospective single-center studies, encompassing a total of 86 patients, indicated that EUS-guided biopsy may achieve a diagnostic yield of 90%, with a median percentage of 82.6% (95% CI: 74.6–90.6%) [22,23]. Furthermore, LP carries an elevated risk of peritoneal metastasis (75.2%). Diagnostic laparoscopy, as well as lavage fluid cytology, may yield false-negative results in up to 35% of cases [5,8]. Irino et al., in their prospective cohort study, observed that among 91 patients suffering from Scirrhous cancer, only 33 were diagnosed using laparoscopy and 38 were diagnosed using lavage fluid cytology, resulting in a combined diagnostic yield for peritoneal disease of 53.8% [24]. Lavage fluid cytology may enhance the diagnosis of peritoneal disease, especially in cases where diagnostic laparoscopy yields negative results. Out of 157 LP patients, only 29 had positive cytology results (18.5%, *p* < 0.001) [8]. Taking everything into account, the demand for diagnostic biomarkers is rapidly increasing. One of the first biomarkers used for LP was introduced by Ichikawa et al. The proposed biomarker was elevated levels of trypsinogen. Increased trypsinogen levels have been linked to positive lymph nodes, liver metastasis and poorly differentiated adenocarcinoma [7]. Miki et al. conducted a comprehensive study involving long non-coding RNA (lnc-RNA), mi-RNA and mRNA, as well as proteins and metabolites involved in the “crosstalk” between cancerous and stromal cells. They found that Mi-RNA-143 levels were increased in scirrhous cancer and correlated with the expression of collagen type 3, through the TGF-b/SMAD pathway, leading to extended fibrosis [25]. Among 27 formalin-fixed paraffin-embedded samples of scirrhous gastric cancer (SGC), 48.1% exhibited the high expression of mi-RNA-143 (13 cases), compared to 24.4% in non-scirrhous gastric cancer cases [25]. Furthermore, Mi-RNA21-5p levels were increased in patients with serous layer infiltration and peritoneal metastasis (T3 and T4). A total of 17 out of 25 patients recorded had a high expression of mi-RNA21-5p, with a significance of *p* = 0.015 [26]. Furthermore, CD-9 expression was found to be increased in exosomes derived from cancer-associated fibroblasts in OCUM-12 cells and NUGC-3 cell lines, and thus is correlated with the increased migration and invasion of cancerous cells [10,18,27]. CD-9 expression was observed in cancerous cells at a rate of 33.3%, in CAFs at 38.1% (32 cases) and combined at 33.3% (28 cases) in patients with scirrhous cancer [27]. However, in other types of GC, the combined appearance of CD-9 reaches up to 14.8%. White blood cells (WBC) may also serve as biomarkers; Pernot et al. observed in their study that LP had a decreased number of natural killer WBC (NK) and T-regulatory cells (T-reg) (6.3% and 3.3%, respectively, out of 27 patients with DGC) as well as FoxP3+ cells (*p* = 0.009) compared to those with the intestinal type of GC (11.5% and 5.2%, respectively) [28]. The advancements in understanding the genetic and epigenetic characteristics of LP will shift the focus of future studies towards liquid biopsy and mi-RNA for a more effective and rapid diagnosis.

## 5. Novel Targets—Approaches

Transmembrane and secretory proteins, particularly those expressed in tumors, can serve as excellent biomarkers for detecting cancer. Gene products implicated in the neoplastic process can also be targeted for therapeutic purposes. One of the most important factors contributing to the rapid progression of LP is the crosstalk between cancerous cells and cancer-associated fibroblasts (CAFs). Antibodies targeting the CD-9 or CD-63 antigens of exosomes or si-CD9 RNA have the potential to impede this crosstalk [10,27]. Tranilast, a form of anthranilic acid, interferes with the interaction between fibroblasts and SGC cells by inhibiting the TGF-b/SMAD pathway. As a result, it leads to slower tumor growth and induces apoptosis [10,29]. Moreover, Saito et al. conducted in vitro experiments that demonstrated how Tranilast prevented the spindle fibroblast-like morphology of human mesothelial peritoneal cells, induced by treatment with TGF-b, causing the cells to adopt a rounder shape [29]. Fibroblast growth factors (FGF), as well as their receptors, are potential targets for impeding the advancement of the disease. Bemarituzumab, an IgG1 monoclonal antibody against FGFR26, blocks the attachment of FGF7, 10 and 12. The selective inhibitors of FGFR 1, 2 and 3 and an FGF antagonist indicate promising results. Additionally, TGFbR inhibitors may activate the tumor suppressor capabilities of FGF-b, thereby inhibiting the growth and metastasis of cancerous cells [10]. In Phase 3 of the FIGHT trial, 5 out of 28 patients with FGFR2 overexpression showed a partial response. Moreover, the overexpression of FGF-18 is regulated by its target, miR-590-5p, making it a potential target [10]. The AMPK pathway, IGF2BP3, IGF1R, MUC16, ARHGAP4, Her-2, MET and mTOR signaling from STKI11/LKB1 are also possible targets [3,10,20,21,30]. Specifically, IGF1R has anti-apoptotic action, and ARHGAP4 is directly associated with the down-regulation of E-cadherin and Claudin 1, making them potent targets [16,21]. *BST*-*2* is overexpressed in 35% of *HER-2* negative GC, and IQGAP3 is found in the cell membrane of cancerous cells, rendering them potential targets. The inhibition of *KIF11* leads to mitotic arrest and the apoptosis of cancerous cells [30]. Claudin 18.2 plays an important role in the diffuse pattern of LP, explaining the rationale behind the development of zolbetuximab, a potent IgG1 monoclonal antibody against Claudin 18.2. This chimeric antibody is in phase 2 of clinical trials and has shown promising results [17]. In the FAST study, the addition of zolbetuximab increased the OS to 13.0 months compared to 8.3 months (*p* < 0.0005) [31,32]. Sai et al. observed, in in vitro experiments, that another inhibitor of CD44-IGF1R, linsitinib, leads to increased OS, promoting apoptosis and inhibiting the viability of cancerous cells [21]. Another approach for increasing the OS involves the simultaneous use of pharmacological ascorbate, which, in mouse models, exhibits selective cytotoxicity on cancer cells, along with oxaliplatin, resulting in better outcomes than chemotherapy alone. In a similar pattern, targets in the NAD+ pathway may prove useful in combination with radiotherapy [33]. Future perspectives include controlling immune checkpoint inhibitors, such as those blocking the interaction between PD1 and PD-L1 and utilizing viruses such as G47D, a modified Herpes simplex virus type 1, which has shown encouraging results in mouse models against solid tumors by inhibiting the growth of subcutaneous cancerous cells [33,34].

## 6. Role of Surgery

Diagnosing LP is often challenging due to the nature of carcinogenesis. Once diagnosed, the tumor is considered unresectable in approximately 41% of cases [35]. However, in some cases, when resected, it typically infiltrates the serosal layer and extends beyond it. Current guidelines and practices recommend gastrectomy with resection margins of at least 5 cm for achieving R0 resection along with D2 lymph node dissection (Japanese Gastric Cancer Association, JGCA). In cases where such surgery is not feasible, palliative gastrectomy may be performed in an attempt to manage bleeding and pyloric obstruction [7,11,36,37]. The choice of gastrectomy type depends on the disease stage, size, layer infiltration and involvement of surrounding structures. Gastrectomy can extend the life expectancy of LP patients, providing a median life expectancy ranging from 12 to 16.7 months. Achieving R0 resection margins significantly improves life expectancy to 35.3 months, compared to R1 and R2 margins, which offer only 15.3 months of OS (*p* < 0.001) [12,13]. With the two main options being total and subtotal/partial gastrectomy, the first is far less promising, with a higher readmission rate, 90-day mortality, as well as a lower 1-year OS rate, compared to partial gastrectomy (47% vs. 52%) [12]. Ikoma et al., in their cohort, observed that the survival of LP patients, even after surgery, does not correlate with that of patients with other types of GC, with LP patients having an extremely poor prognosis (21.8 vs. 91.0 months, *p* < 0.001) [8]. While LP patients generally have reduced survival compared to other gastric cancer types, stage stratification reveals that the prognosis of LP patients is similar to that of patients at the same stage of other GC types [7,38]. LP often affects the upper third of the stomach (15–20%), and gastrectomy alone may not be sufficient since LP can metastasize to the splenic hilum lymph node (#10) and the spleen. In such cases, splenic dissection, along with the splenic hilar LN, is considered. This approach can increase overall survival, outweighing the complications associated with spleen resection [11]. As mentioned already, serosal layer infiltration is a significant factor in the tumor’s spread to the surrounding tissues, turning the R0 resection into a difficult endeavor. Xiao et al. retrospectively analyzed the utilization of extended multiorgan resection (EMR) in patients with locally advanced GC. They observed a clear advantage in OS after EMR, compared to palliative gastrectomy, rather than when compared to gastrectomy alone (27 m vs. 11 m vs. 44 m). However, LP was identified as an independent factor for a poor prognosis and the OS after EMR was similar to that following palliative surgery (11 m). Thus, EMR for LP patients should be approached with caution due to the elevated morbidity and low survival [36].

## 7. Discussion

Gastric cancer ranks as the fifth most common malignancy, with approximately 1.1 million new cases reported each year. LP, a rare and highly aggressive subtype of gastric cancer, constitutes 10% of all GC cases [2]. LP exhibits a distinctive development pattern, originating in the fundic glands and spreading throughout the entire stomach, concurrently triggering a desmoplastic reaction. This desmoplastic reaction, combined with fibrosis, leads to the thickening of gastric folds, earning it the moniker “Leather bottle stomach”. In a significant 91.1% of histologically confirmed cases, LP presents as a poorly differentiated adenocarcinoma, while in 77.7% of cases, it is often histologically linked with signet ring cells. LP typically manifests with symptoms such as dyspnea, nausea, vomiting and anorexia, which are general and not conclusive indicators of the disease. The accumulation of ascites may also be observed when peritoneal metastasis is present [15]. Unfortunately, the limited diagnostic yield of current methods, as well as the absence of specific diagnostic criteria for its diagnosis, is a hinderance, often making the tumor inoperable [7,35]. The gold standard for diagnosing LP is endoscopic ultrasound with biopsy, but non-diagnostic biopsies are found in 30–55% of cases [6,7]. The sensitivity and specificity for diagnosing T3 and T4 parietal invasion are 77% and 100%, respectively, while the evaluation for LN is 75% and 80%, respectively [39]. Recent research has suggested the use of a mucosal flap in conjunction with submucosal endoscopic resection. Furthermore, numerous studies have emerged utilizing endoscopic methodologies, such as endocytoscopy and endomicroscopy for the identification of SGC [5]. Alternative diagnostic techniques include endoscopy, upper gastrointestinal contrast study, computed tomography, 18FDG PET and magnetic resonance imaging. The endoscopic appearance of LP can resemble that of gastric lymphoma, Menetrier disease, granulomatous disease or metastasis due the hypertrophic mucosal folds. When these features are absent, LP may present as a non-specific gastritis or normal mucosa and the “flat type” of LP can be confused with atrophic gastritis [6,7]. Chicoteau et al., in their retrospective study, validated a pre-therapeutic diagnostic score for LP known as the Saint Louis Linitis Score. These criteria have been transformed into a universal diagnostic tool with a sensitivity and specificity of 94% and 88.7%, respectively. The endoscopic criteria include the presence of large folds and/or parietal thickening on at least one segment, pangastric infiltration and gastric stenosis observed during endoscopy, circumferential thickening on at least one segment, the presence of the third hyperechogenic layer on endoscopic ultrasound and the identification of signet ring cells [40]. In diagnosing positive LN or parietal involvement in LP, CT demonstrates similar sensitivity and specificity to gastric adenocarcinoma. However, CT exhibits reduced sensitivity when it comes to detecting peritoneal metastasis [41].

The etiology of LP is not solely rooted in mutations within cancerous cells, it also hinges on the interactions between stromal and cancerous cells. Huang et al. assert that these mutations result from spontaneous deaminations or defects in DNA mismatch repair mechanisms, mostly occurring after failures in repairing DNA double-strand breaks [9]. LP stands out as one of the most heterogeneous cancers in terms of mutated genes, but in each distinct case, it exhibits the same mutations across all its regions, with a prevalence ranging from 68–95% [9]. The invasive nature of LP is attributed to mutations in genes and proteins associated with tight junctions. Liu et al. observed that the *CDH1* gene, responsible for encoding the classic cadherin molecule, is mutated in approximately 20% of LP patients and 25% of patients with DGC. The down-regulation of E-cadherin may result from the down-regulation of miRNA-200c, which influences the expression of ZEB1. MiRNA-200c belongs to the miRNA-200 family, which regulates the epithelial–mesenchymal transition [2,3]. Claudin 18.2 is another molecule crucial for the functioning of tight junctions. Mutations in this molecule impair its function, thereby promoting tumor growth, proliferation, invasion and metastasis. Zolbetuximab, a highly potent chimeric IgG1 mAb specific for Claudin 18.2, has shown promising results in the phase 3 SPOTLIGHT clinical trial, as evidenced by increased cancer-free survival and OS rates [17,42]. *ARHGAP4* is a gene that does not directly affect tight junctions but is associated with the increased invasion and migration of cancerous cells. This is attributed to the increased expression of MMP2 and 3 and the down-regulation of E-cadherin and Claudin 1. *NOS-3*, another oncogene linked to diffuse cancer and LN metastasis as well as peritoneal metastasis, plays an important role in the epithelial–mesenchymal transition (EMT) through MAPK signaling [16]. These genes can negatively impact OS and recurrence-free survival. Crucial to LP tumorigenesis are the tumor suppressor HIPPO pathway and the PI3K-AKT pathway. Among the primarily mutated genes in the HIPPO pathway are *FAT3* and *PTPN14*, while IGTA/IGTB integrins belong to the superfamily. All of those genes are up-regulated in LP due to the activation of the PI3K-AKT pathway. A deeper comprehension of these pathways and the molecules comprising them will enhance our understanding of LP [3]. GLP also shares the same mutated genes with DGC, including *CDH1, RHOA* and *MUC6*.

Mutations in stromal cells are of equal importance, as they profoundly impact the invasion, metastasis and growth of cancerous cells. They also influence processes like EMT, angiogenesis and extracellular matrix remodeling [43]. Cancerous cells engage in interactions with cancer-associated fibroblasts, which make up as much as 90% of the tumor. CAFs are a diverse group of cells that can originate from local fibroblasts, bone marrow precursors, mesenchymal cells and pericytes that undergo EMT [7,43]. The communication between CAFs and cancerous cells is facilitated by exosomes, including CD-63, which can lead to scirrhous cancer. This interaction results in increased tumor depth, size, LN metastasis and CD-9 expression. Miki et al. observed that CAF-secreted exosomes exhibit higher CD9 expression compared to exosomes from normal fibroblasts. Furthermore, CAF-secreted exosomes are selectively taken up by SGC cells, not by other types of GC cells. Exosomes from CAFs enhance SGC cell migration and invasion, an effect that can be blocked by antibodies or siRNA targeting exosomal CD9 [10,18]. Stromal cells also produce HGF, which has a paracrine function in cancerous cells. HGF increases their survival and proliferation, while simultaneously promoting fibrosis and diffuse infiltration, leading to peritoneal carcinomatosis and ascites [14]. These effects are mediated by the activation of the MET pathway. The inhibitors of this pathway can reduce ascites accumulation in mice over a 10-day period. When combined with a VEGFR2 inhibitor, like cabozantinib, the results can persist for the duration of the treatment [14]. CAFs are also responsible for secreting FGF, which stimulates tumor growth and the desmoplastic reaction observed in 40.4% of LP cases compared to other types of GC [10]. SGC predominantly comprises collagen type IV. Antifibrotic medications, such as losartan, tranilast and hyaluronidase, have the potential to increase tumor perfusion, enhance medication delivery and promote immune cell infiltration [9]. Intraperitoneal chemotherapy is also being investigated as an alternative to conventional IV chemotherapy for peritoneal carcinomatosis. Taxane-based IPC takes advantage of the antifibrotic properties of the agent [1]. Adipocytes found around tumors, along their borders, or within the tumor itself, serve a function similar to CAFs. These are referred to as cancer-associated adipocytes (CAAs) and they promote angiogenesis, cancerous cell infiltration and serve as an initial step for peritoneal infiltration [44]. Saito et al. elucidated the mechanisms underlying the interaction between cancerous cells and human peritoneal mesothelial cells (HPMCs) in the development of fibrosis and metastasis in GC caused by TGF-b1. TGF-b1 transforms these cells into spindle-shaped cells and reduces the expression of E-cadherin, facilitating the infiltration and metastasis of cancerous cells [29].

The prognosis for LP is generally poor and the OS is limited. Ikoma et al. have reported that the 1-, 2- and 5-year survival rates without surgery are 59%, 12% and 5%, respectively, while after surgery, these rates improve to 69%, 27% and 18%, respectively, for LP patients. These rates are significantly lower when compared to non-LP patients [8]. The median OS varies widely, ranging from 3.6 to 33.5 months. Ayub et al. observed that the 5-year survival rate after surgery can reach up to 29% and post-surgery OS is not significantly different from that of patients with DGC [12,13]. Recurrences are frequent, even after curative surgery, often occurring locally or in the peritoneal region through peritoneal seeding. The highest incidence of recurrences, at 60–70%, is observed at 18 months after surgery [45]. Key prognostic factors include tumor size (>8 cm) and gastric bare area infiltration, which significantly diminishes patient survival (*p* < 0.001). The latter is responsible for 36% of retroperitoneal infiltration and 21.3% of LN metastasis, observed in 11.3% of the patients. However, a prognosis is improved when signet ring cells make up more than 50% of the tumor composition [5,44,46].

The standard treatment of LP typically involves gastrectomy with resection margins exceeding 5 cm to achieve R0 resection and D2 lymph node dissection following the guidelines of the Japanese Gastric Cancer Association. In cases where this is not feasible, palliative gastrectomy is performed [7,11,36,37]. Radiotherapy has limited effectiveness against diffuse cancers [7]. The use of chemotherapy, especially in the neoadjuvant setting, has seen advancements in recent years. Today, various chemotherapeutic regimens have shown positive results in neoadjuvant (NAC) and intraperitoneal chemotherapy (IP). The IP administration of Paclitaxel (PTX) or Docetaxel has been found to improve median survival by more than 20 months and increase 1-year OS in up to 70% of patients with peritoneal disease [1]. Kodera et al. have demonstrated the safety of the IP administration of PTX compared with the IV administration [47]. Nonetheless, Takayashi et al. did not detect any significant benefits of PTX IP administration compared to IV administration, and the PHOENIX-GC2 trial aims to provide more conclusive evidence regarding this regimen [48,49]. Song et al. observed that postoperative chemotherapy/radiotherapy benefits patients with stage 3 disease and tumor sizes over 8 cm. One of the main regimens for postoperative chemotherapy is FLOT, consisting of 5-FU, leucovorin, oxaliplatin and docetaxel, which has shown benefits for DGC [45]. The DCS neoadjuvant regimen, comprising docetaxel, cisplatin and S-1, has increased the 3-year survival rate up to 71.9%, compared to cisplatin and S-1 alone [50,51]. The KD0G1001 trial, conducted by Hosoda, achieved similar results, enhancing OS to 77.5% and enabling R0 resection in 90% of cases [52,53]. Moreover, a phase 2 trial, testing the efficacy of the preoperative IV administration of Epirubicin, etoposide, oxaliplatin and oral S-1 for LP patients, has shown promising results with a response rate of 55.6%, a disease control rate of 69.4% and an R0 resection rate of 66.7%. The median OS was 27.1 months [54].

For patients with peritoneal metastases, hyperthermic intraperitoneal chemoperfusion (HIPEC) is becoming an increasingly popular treatment option. The adoption of HIPEC as a prophylactic measure for patients at high risk of macroscopic peritoneal seeding may be a promising strategy. The PREVENT study, a phase 3 trial, aims to establish HIPEC as a standard practice [45].

## 8. Conclusions

Linitis Plastica is an exceptionally heterogeneous tumor, and comprehending its molecular underpinnings will provide the groundwork for innovative treatments, diagnostic approaches and personalized medicine. Surgery remains pivotal for enhancing the survival prospects of patients, and the integration of adjuvant or IP postoperative chemotherapy can proliferate patients’ survival. Future research should prioritize unraveling the mechanisms behind tumorigenesis and the development of novel therapeutic agents to effectively manage this disease.

## Figures and Tables

**Table 1 ijms-24-14680-t001:** Main oncogenes in Linitis Plastica.

Genes	Role of the Genes	Frequency
*Angiopoietin-like 4 (ANGPTL4)*	Inhibits endothelial cell migration, angiogenesis, regulatory role in lipid metabolism, metastasis.	
*AOX1*	Encoded aldehyde oxidase—oncogenic role	30%
*ARHGAP4*	Increases the expression of matrix metalloproteinase 2 and 9 and decreases E-cadherin and Claudin 1	
*ARID1A*	Driver gene	0–40%
*ASPN*	Oncogenic driver or a tumor suppressor gene	
*CD44-IGF1R* fusion gene and *BORCS5-ETV6* fusion		
*CD-63*	Correlated with SRC, lymph node metastasis, peritoneal metastasis and tumor progression	
*CDH1*	Encodes E-cadherin	20–65%
Claudin 18.2	Impairs tight junctions	14.1% in advanced gastric cancers
*C-met* gene	HGF receptor	
*c-MYC*		
*DCDC1, GOLGA6L10*, and *MUC12)*		65%
Desmoglein-2	Down-regulated in diffuse gastric cancer	
*ELK4, MDM4, SLC45A3, H3F3A* and *ZNF429.*		
*FAK/Src/PI3K-Akt/ERK*	Metastasis	
*FAT3*, and *PTPN14* from the *HIPPO* pathway	Tumor suppressor pathway	75%
*FGFR2*	Increase invasion and migration through THBS-1 pathway	>5–10%
*MET*	Tyrosine kinase receptor for HGF	
*MMP 7*	Degrade the extracellular matrix and also promotes apoptosis evasion in cancer cells	
*MUC6*		20%
*NOS3*	Enhance migration and invasion and peritoneal dissemination	
*OR51B5*	Increases cell survival and collagen biosynthesis	
*p-27*	Cell cycle inhibitor	
*RELN*	Driver gene	75%
*STK11/LKB1*	Tumor suppressor gene	
*TP53/TTN*

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
