# Peer review of "The Genomic Signatures of Linitis Plastica Signal the Entrance into a New Era: Novel Approaches for Diagnosis and Treatment"

_ijms, 2023, doi:10.3390/ijms241914680_

Round 1

Reviewer 1 Report

The authors have written a review article on linitis plastica (LP) that describes the molecular changes found in LP, biomarkers and novel target approaches. The topic is of interest, but the manuscript needs revision. The main impression is that a more stringent structure, correction of errors and inconsistencies and proofreading is necessary.

The title of the manuscript does not reflect the contents of the manuscript so well and could be modified. The Discussion is unfocused and difficult to follow and several of the first sentences in the Discussion seem to be copied from the introduction.   

There are many minor orthographic errors that gives the manuscript an unpolished appearance and the general impression would improve if these were corrected. Look for spaces, comma vs period, many sporadic or random capital letters.

Please use for instance the LP abbreviation consistently after being introduced on page 1. Similarly for all other abbreviations, spell out if appropriate, and use the abbreviation downstream.

Page 2: “Defining is the role of stromal…” is incoherent.

In the Methods section: please define old studies that were excluded. Cut-off?

Table1: please structure the oncogenes either alphabetically or preferably ranked after frequency with a reference.

Page 3: “lead to Gastric Linitis Plastica”. Please delete “gastric” which seems obvious or clarify.

Page 4: The section headline Biomarkers also cover diagnostic procedures, which is normally not considered a biomarker. Please adjust the headline or the contents below the headline.

Page 4: please define US.

Page 4: Please support the statement that false-negative biopsies are 30-55% and the number of biopsies is here important. This statement is repeated on page 6, but seems here to be related to EUS-guided biopsies.

Page 4. When diagnostic laparoscopy was false negative in 35% of patients, was cytological examination of lavage fluid performed in references 5 and 8? Please specify and search for publications reporting the results from examination of lavage cytology.

When describing the performance of biomarkers in section 4, please quote numbers describing the performance of such biomarkers.

In section 5, please quote numbers describing the performance of the best novel drugs on survival. This will provide useful perspective on the status of the research field. If some findings are from in vitro studies only, please specify this.

Page 5: It is stated that in linitis plastica that is resectable the cancer extends over the serosa. Is this always the case? Please quote a frequency with reference.

Page 6: please define OS and GC.

Please see above. Thorough proofreading desirable. 

Author Response

Dear Reviewer, thank you for your revisions. The title has now been changed, the entire review has been revised, orthographic errors have been corrected, and we have undergone extensive proofreading.

Following the initial introduction of the abbreviation 'Linitis Plastica' the same abbreviation LP is consistently used throughout the document.

On page 2, we have clarified the meaning of the sentence, 'Defining is the role of stromal…' into “Determining is the role of stromal mutations and the influence of stromal cells to the cancerous ones”

In accordance with the exclusion criteria outlined in the methods section, articles published prior to 2017 have been excluded, with the exception of two important articles that were utilized for providing general information rather than for incorporating up-to-date developments in the field of LP.

The Genes are categorized alphabetically.

On page 3, we have removed the word 'Gastric,' and on page 4, we have defined the abbreviation 'US' as Ultrasound.

In the Biomarkers section of the review, the mention of diagnostic techniques was intended to emphasize the limited diagnostic efficacy of traditional methods and to underscore the necessity for a functional biomarker. Therefore, we did not delve into a comprehensive analysis of the entire spectrum of diagnostic procedures, their sensitivity, specificity, and changing the title of this section would be misleading.

Within the Biomarkers section, we have provided additional information about cases that underwent EUS biopsy, as well as those that underwent diagnostic laparoscopy and lavage fluid examination:

“A recent meta-analysis of 4 Retrospective single-center studies, of 86 patients in total, indicated that EUS-guided biopsy may lead to a diagnostic yield of 90%, with a median percentage of 82.6% (95% CI: 74.6–90.6%)” and “Irino et al, in their prospective cohort study, observed that out of 91 patients suffering Scirrhous cancer, only 33 were diagnosed with laparoscopy and 38 were diagnosed with lavage fluid cytology, giving a combined diagnostic yield for peritoneal disease of 53,8% [49]. Lavage fluid cytology may increase the diagnosis of peritoneal disease, in cases with negative diagnostic laparoscopy. Out of 157 patients with LP, only 29 of them had a positive cytology (18,5%, p<0.001)”.

We have also included four additional citations to enhance the comprehensiveness of the text. Unfortunately, specific information regarding the number of biopsies per patient was not available.

In sections 4 and 5, when describing the performance of biomarkers and novel drugs, we have included numerical data to support their utility, performance, and their impact on Overall Survival. We have also indicated which of these results are based on in vitro studies.

Regarding the resectability of LP after the infiltration of the serosal layer, there was a typographical error in the previous version. The correct phrasing should be, 'However, in cases where it is resected, it infiltrates the serosal layer and extends beyond it.'

Finally, for clarity, we have defined the abbreviations 'OS' as 'Overall Survival',  'GC' as 'Gastric Cancer' and we add a glossary below the abstract containing all the abbreviations used.

Reviewer 2 Report

This review paper by Christodoulidis et al. described the origin and research progress of LP, a subtype of gastric cancer, characterized by its great heterogeneity of mutational profiles. It is well-written and could serve as a valuable resource to understand the key questions researchers have about LP. 

The main issue for this manuscript is with the summary of biomarkers. The authors should specify which dataset(s) they used to summarize the frequency of mutations or copy number changes. The main reason is that if the authors used multiple datasets, due to the mutation filtration difference (number of reads, allele frequencies, etc.), the frequency of mutation might not be accurate unless the authors used a uniform threshold. 

Minor comment: is it possible to also show some visualizations of oncoplots or heatmaps to show the heterogeneity for LP? Or maybe some expression analysis to see if beyond the heterogeneity, are there a gene set that are upregulated? 

Author Response

Dear reviewer, thank you very much for your constructive feedback. Regarding your comment about the dataset we used, no specific dataset was employed since this is a review rather than a meta-analysis. Therefore, a straightforward reference to the genes was made, without statistical or stratified analysis. For the same reason, the addition of an oncoplot or heatmap is not feasible. However, for better organization of the genes, we present them alphabetically in the table. Finally, the following changes were implemented:

Within the Biomarkers section, we have provided additional information about cases that underwent EUS biopsy, as well as those that underwent diagnostic laparoscopy and lavage fluid examination:

“A recent meta-analysis of 4 Retrospective single-center studies, of 86 patients in total, indicated that EUS-guided biopsy may lead to a diagnostic yield of 90%, with a median percentage of 82.6% (95% CI: 74.6–90.6%)” and “Irino et al, in their prospective cohort study, observed that out of 91 patients suffering Scirrhous cancer, only 33 were diagnosed with laparoscopy and 38 were diagnosed with lavage fluid cytology, giving a combined diagnostic yield for peritoneal disease of 53,8% [49]. Lavage fluid cytology may increase the diagnosis of peritoneal disease, in cases with negative diagnostic laparoscopy. Out of 157 patients with LP, only 29 of them had a positive cytology (18,5%, p<0.001)”.

In sections 4 and 5, when describing the performance of biomarkers and novel drugs, we have included numerical data to support their utility, performance, and their impact on Overall Survival. 

Finally, for clarity, we add a glossary below the abstract containing all the abbreviations used.

Round 2

Reviewer 1 Report

I thank the authors for revising the manuscript and the contents merit publication. I would still recommend the manuscript to be proof-read by a native English speaker. The new title would probably not pass proof-reading and the last part could be changes to "Novel approaches for diagnosis and treatment". 

Please note that this is just one example and that detailed language revision is beyond the responsibility of a reviewer. 

Please see above.

Author Response

Thank you for your constructive comments. The title of the review has been changed to 'The Genomic Signatures of Linitis Plastica signal the entrance into a new Era. Novel approaches for Diagnosis and Treatment.'

The whole manuscript has undergone extensive proofreading and detailed language revisions by a native english speaker. The changes are highlighted.